# Learning to Make Analogies by Contrasting Abstract Relational Structure

**Felix Hill\***, **Adam Santoro,\*** **David G. T. Barrett, Ari Morcos & Tim Lillicrap**
Deepmind, London
`{felixhill,adamsantoro,barrettdavid,arimorcos,countzero}@google.com`

## Abstract

Analogical reasoning has been a principal focus of various waves of AI research. Analogy is particularly challenging for machines because it requires relational structures to be represented such that they can be flexibly applied across diverse domains of experience. Here, we study how analogical reasoning can be induced in neural networks that learn to perceive and reason about raw visual data. We find that the critical factor for inducing such a capacity is not an elaborate architecture, but rather, careful attention to the choice of data and the manner in which it is presented to the model. The most robust capacity for analogical reasoning is induced when networks learn analogies by contrasting abstract relational structures in their input domains, a training method that uses only the input data to force models to learn about important abstract features. Using this technique we demonstrate capacities for complex, visual and symbolic analogy making and generalisation in even the simplest neural network architectures.

## 1 Introduction

The ability to make analogies – that is, to flexibly map familiar relations from one domain of experience to another – is a fundamental ingredient of human intelligence and creativity (Gentner, 1983; Hofstadter, 1996; Hummel & Holyoak, 1997; Lovett & Forbus, 2017). As noted, for instance, by Holyoak & Thagard (1995), analogies gave Roman scientists a deeper understanding of sound when they leveraged knowledge of a familiar source domain (water waves in the sea) to better understand the structure of an unfamiliar target domain (acoustics). The Romans 'aligned' relational principles about water waves (periodicity, bending round corners, rebounding off solids) to phenomena observed in acoustics, in spite of the numerous perceptual and physical differences between water and sound. This flexible alignment, or mapping, of relational structure between source and target domains, independent of perceptual congruence, is a prototypical example of analogy making.

It has proven particularly challenging to replicate processes of analogical thought in machines. Many classical or symbolic AI models lack the flexibility to apply predicates or operations across diverse domains, particularly those that may have never previously been observed. It is natural to consider, however, whether the strengths of modern neural network-based models can be exploited to solve difficult analogical problems, given their capacity to represent stimuli at different levels of abstraction and to enable flexible, context-dependent computation over noisy and ambiguous inputs.

In this work we demonstrate that well-known neural network architectures can indeed learn to make analogies with remarkable generality and flexibility. This ability, however, is critically dependent on a method of training we call *learning analogies by contrasting abstract relational structure* (LABC). We show that simple architectures can be trained using this approach to apply abstract relations to never-before-seen source-target domain mappings, and even to entirely unfamiliar target domains.

Our work differs from previous computational models of analogy in two important ways. First, unlike previous neural network models of analogy, we optimize a single model to perform both stimulus representation and cross-domain mapping jointly. This allows us to explore the potential benefit of interactions between perception, representation and inter-domain alignment, a question of some debate in the analogy literature (Forbus et al., 1998). Second, we do not instantiate an

---

*Equal contribution

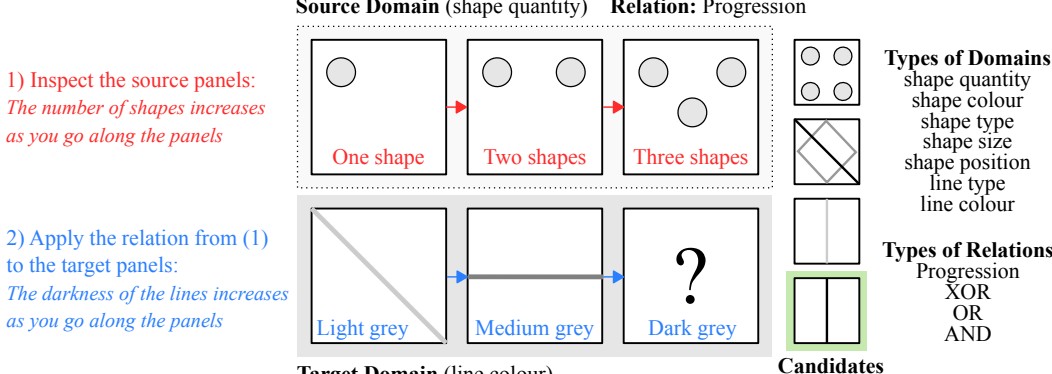

Figure 1: **A visual analogy problem.** In this example, the model must (1) identify a relation (Progression ) on a particular domain ( `shape quantity` ) in the source sequence (top), and (2) apply it to a different domain ( `line color` ) in order to find the candidate answer panel that correctly completes target sequence (bottom). There are seven possible domains and four possible relations in the dataset.

explicit cognitive theory or analogy-like computation in our model architecture, but instead use this theoretical insight to inform the way in which the model is trained.

## 2 ANALOGIES AS HIGH-LEVEL PERCEPTION AND STRUCTURE MAPPING

Perhaps the best-known explanation of human analogical reasoning is Structure Mapping Theory (SMT) (Gentner, 1983). SMT emphasizes the distinction between two means of comparing domains of experience; analogy and similarity. According to SMT, two domains are similar if they share many *attributes* (i.e. properties that can be expressed with a one-place predicate like `BLUE(sea)` ), whereas they are analogous if they share few attributes but many relations (i.e. properties expressed by many-place predicates like `BENDS-AROUND(sea, solid-objects)` ). SMT assumes that our perceptions can be represented as collections of attributes and structured relations, and that these representations do not necessarily depend on the subsequent mappings that use them.

The High-Level Perception (HLP) theory of analogy (Chalmers et al., 1992; Mitchell, 1993) instead construes analogy as a function of tightly-interacting perceptual and reasoning processes, positing that the creation of stimulus representations and the alignment of those representations are mutually dependent. For example, when making an analogy between the sea and acoustics, we might represent certain perceptual features (the fact that the sea appears to be moving) and ignore others (the fact that the sea looks blue), because the particular comparison that we make depends on location and direction, and not on colour.

In this work we aim to induce flexible analogy making in neural networks by drawing inspiration from both SMT and HLP. The perceptual and domain-comparison components of our models are connected and jointly optimised end-to-end, which, as posited by HLP, reflects a high degree of interaction between perception and domain comparison. On the other hand, the key insight of this paper, LABC, is directly motivated by SMT. We find that LABC greatly enhances the ability of our networks to resolve analogies in a generalisable way by encouraging them to compare inputs at the more abstract level of relations rather than the less abstract level of attributes. LABC organizes the training data such that the inference and mapping of relational structure is essential for good performance. This means that problems cannot be resolved by considering mere similarity of attributes, or even less appropriately, via spurious surface-level statistics or memorization.

## 3 VISUAL ANALOGY PROBLEMS

Our first experiments involve greyscale visual scenes similar to those previously applied to test both human reasoning ability (Raven, 1983; Geary et al., 2000) and reasoning in machine learning models (Bongard, 1967; Fleuret et al., 2011; Barrett et al., 2018). Each scene is composed of a *source sequence*, consisting of three *panels* (distinct images), a target sequence, consisting of two panels, and four candidate answer panels (Fig. 1). In the source sequence a relation $r$ is instantiated, where $r$ is one of four possible relations from the set $R = \{$ XOR , OR , AND , Progression $\}$. Models must then consider the two panels in the target sequence, together with the four candidate answer panels, to determine which answer panel best completes the target sequence – by analogy with the source sequence – so that $r$ is also instantiated (Fig. 2).

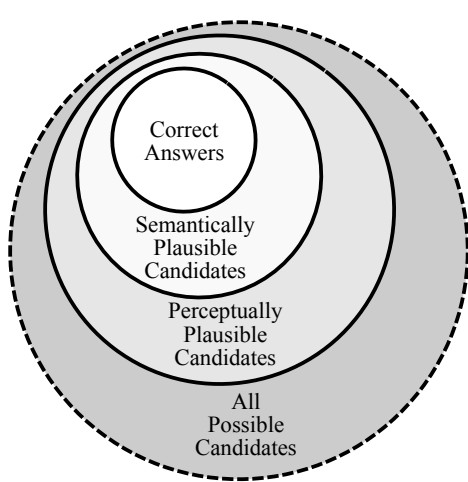

Figure 2: In LABC, the multiple choice candidates are all semantically plausible, in that they are all consistent completions of the target domain using some relation. Only the correct answer uses the same relation as the source domain, so, the only way to solve the problem is to use analogical reasoning. In contrast, perceptually plausible incorrect candidates are consistent with the target domain attributes but not the relations and all other possible candidates are inconsistent with the target domain relation and attributes.

The notion of *domain* is critical to analogy. In our visual analogy task, a relation can be instantiated in one of seven different domains: line type , line colour , shape type , shape colour , shape size , shape quantity and shape position (see Fig. 1 and Appendix Fig. 7 for examples). Within a panel of a given domain, the attributes present in the scene (such as the colour of the shapes or the positions of the lines) can take one of $10$ possible *values*. A question in the dataset is therefore defined by a relation $r$, a domain $d_s$ on which $r$ is instantiated in the source sequence, a set of values for the source-domain $v_1^s \cdots v_3^s$, a target domain $d_t$, values for the target-domain $v_1^t \cdots v_3^t$, the position $k \in \{1 \cdots 4\}$ of the correct answer among the answer candidate panels and whatever is instantiated in the three incorrect candidate answer panels $c_i, i \neq k$. Note, however, that the values of certain domain attributes that are not relevant to a given question (such as the colour of shapes in the shape quantity domain) still have to be selected, and can vary freely. Thus, despite the small number of latent factors involved, the space of possible questions is of the order of ten million.

The interplay between relations, domains and values makes it possible to construct questions that require increasing degrees of abstraction and analogy-making. The simplest case involves a relation $r$, a domain $d_s == d_t$, and values $v_i^s == v_i^t$ that are common to both source and target sequences (Fig 3a). To solve such a question a model must identify a candidate answer panel that results in a copy of the source sequence in the target sequence. This does not seem to require any meaningful understanding of $r$, nor any particular analogy-making. Somewhat greater abstraction and analogy-making is required for questions involving a single domain ($d_s == d_t$), but different values in the source and target sequence $v_i^s \neq v_i^t$ (Fig 3b). In this case the model must learn that, in a given domain, the relation $r$ can apply to a range of different values. Finally, in the full analogy questions considered in this study, the relation $r$ can be instantiated on different domains in the source and target sequences (i.e. $d_t \neq d_s$; Fig 3c). These questions require a sensitivity to the idea that a single relation $r$ can be applied in different (but related) ways to different domains of experience.[1]

---

[1]The visual analogy dataset can be downloaded from https://github.com/deepmind/abstract-reasoning-matrices

### 3.1 METHODS

Our model consisted of a simple perceptual front-end – a convolutional neural network (CNN) – which provided input for a recurrent neural network (RNN) by producing embeddings for each image panel independently. The RNN processed the source sequence embeddings, the target sequence embeddings, and a *single* candidate embedding, to produce a scalar score. Four such passes (one for each source-target-candidate set) produced four scalar scores, quantifying how the model evaluated the suitability of the particular candidate. Finally, a softmax was computed across the scores to select the model's 'answer'. Further model details are in appendix 7.1.

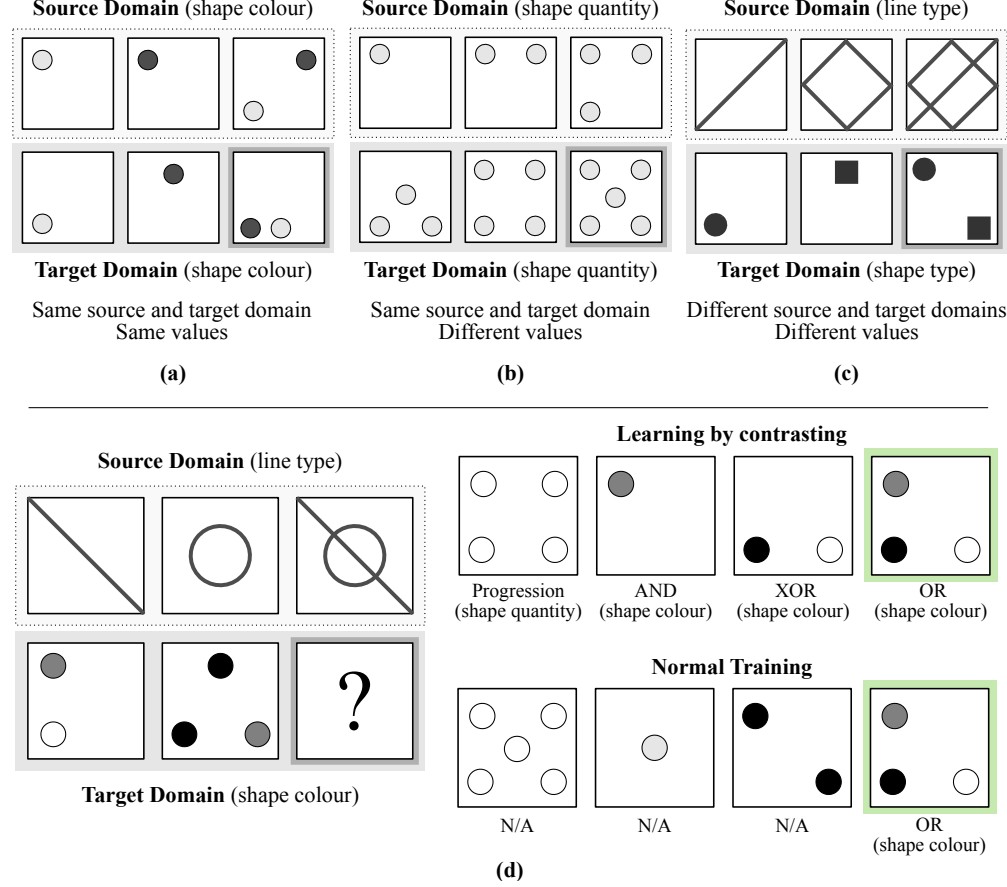

Figure 3: **(a), (b), (c) Three types of visual reasoning questions.** Each question requires a different degree of analogy making, with the question on the right demanding the most fluid and abstract application of the underlying relation. **(d) Learning analogies by contrasting.** When learning by contrasting, each answer choice is consistent with a relational structure in the target sequence. Only the correct answer choice is consistent with relations in both the source and target domains. This forces the network to consider the source sequence to infer the correct structure.

**Learning Analogies By Contrasting (LABC)** In the default setting of our data generator – the *normal training* regime – for a question involving source domain $d_s$, target domain $d_t$ and relation $r$, the candidate answers can contain any (incorrect) values chosen at random from $d_t$ (Fig. 2). By selecting incorrect candidate answers from the same domain $d_t$ as the correct answer, we ensure that they are perceptually plausible, so that the problem cannot be solved trivially by matching the domain of the question to one of the answers. Even so, the baseline training regime may allow models to find perceptual correlations that allow it to arrive at the correct answer consistently over the training data. We can make this less likely by instead training the model to contrast abstract relational structure – *the LABC regime* – simply by ensuring that incorrect answers are both perceptually *and* semantically plausible (Fig. 2). More specifically, incorrect answers are selected from $d_t$ such that each answer

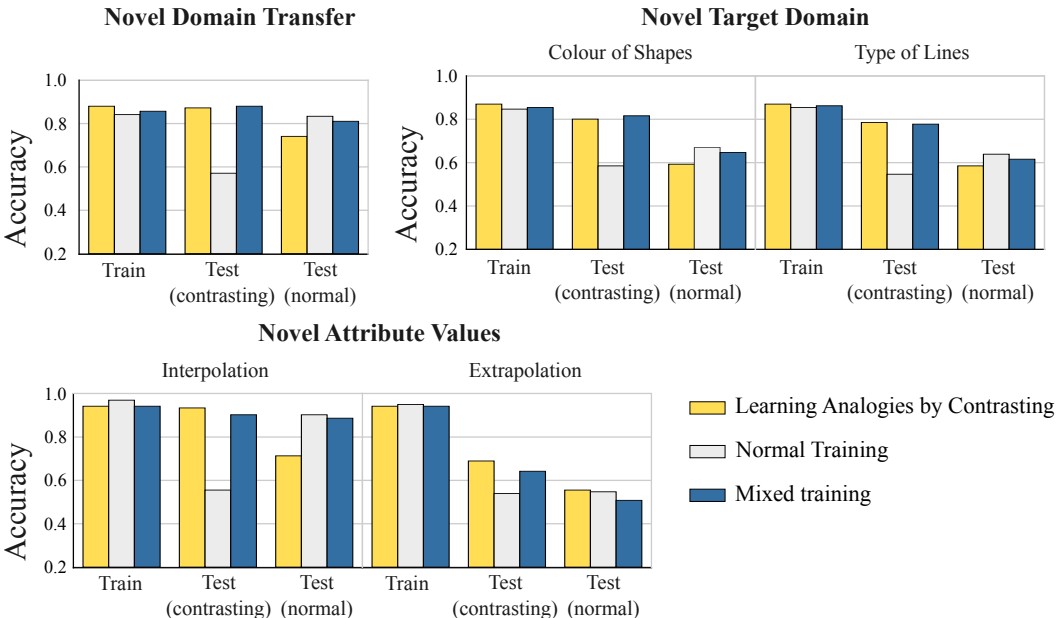

Figure 4: **Results of the three experiments in the visual analogy domain for a network that learns from random candidate answers, by contrasting abstract structures or both types of question interleaved.** Bar heights depict the means across eight seeds in each condition; standard errors were $< 0.01$ for each condition (not shown – see the appendix Table 4 for the values)

$c_i$ completes a decoy relation $\hat{r}_i \neq r$ with the target sequence. LABC ensures that, during training, models have no alternative but to first observe a relation $r$ in the source domain and consider and complete the same relation in the target domain - i.e. to execute a full analogical reasoning step.[2]

Note that in all experiments reported below, we generated 600,000 training questions, 10,000 validation questions and test sets of 100,000 questions. These data will be published with the paper.

### 3.2 EXPERIMENT 1: NOVEL DOMAIN TRANSFER

A key aspect of analogy-making is the process of comparing or aligning two domains. We can measure how well models acquire this ability by testing them on analogies involving unfamiliar source domain $\rightarrow$ target domain transfers. For each of the seven possible target domains $d_t$ we randomly selected a source domain $d_s \neq d_t$, yielding a test set of seven domain transfer pairs $[d_s \rightarrow d_t]$. Our models were then trained on questions involving one of the remaining $7 \times 7 - 7 = 42$ domain transfer pairs. For a test question involving domains $d_s$ and $d_t$, each model was therefore familiar with $d_s$ and $d_t$ but had not been trained to make an analogy from $d_s$ to $d_t$.

We found that a network can indeed learn to apply relations by analogy involving novel domain transfers, but that this ability crucially relies on learning by contrasting. The effect is strong; for the most focused test questions involving semantically plausible (contrasting) candidate answers the model trained by contrasting achieves 83% accuracy (depending on the held-out domain), versus 58% for a model trained with randomly-chosen candidate answers.

### 3.3 EXPERIMENT 2: NOVEL TARGET DOMAIN

Humans can use analogies to better understand comparatively unfamiliar domains, as in the Roman explanation of acoustics by analogy with the sea. To capture this scenario, we held out two domains ( `line type` and `shape colour` , chosen at random) from the model's training data, and ensured

---

[2]It is important to note that that LABC as described here relies on our understanding of the underlying data-generating process; we demonstrate its application that does not require such understanding in Sec. 5.3.

|  | **LABC training** | **Normal training** |
|---|---|---|
| RNN (this paper) | **0.90** (0.01) | 0.79 (0.09) |
| ResNet-50 | **0.82** (0.02) | 0.77 (0.10) |
| Parallel ResNet-50 | **0.89** (0.01) | 0.75 (0.14) |
| Parallel Relation Net (WReN) | **0.95** (0.017) | 0.72 (0.21) |

Table 1: Test performance of a selection of visual reasoning network architectures trained in the normal regime and with LABC on the **Novel Domain Transfer** experiment. The test questions include those with both semantically-plausible and merely perceptually-plausible incorrect answers. Mean (+SD) for 10 models in each condition with different random initialisations.

that each test question involved one of these domains. To make sense of the test questions, a model must therefore (presumably) learn to represent the relations in the dataset in a sufficiently general way that this knowledge can be applied to completely novel domains. Any model that resolves such problems successfully must therefore exploit any (rudimentary) perceptual similarity between the test domain and the domains that were observed during training. For instance, the process of applying a relation in the `shape colour` domain may recruit similar feature detectors to those required when applying it to the `line colour` domain.

Surprisingly, we found that the network can indeed learn to make sense of the unfamiliar target domains in the test questions, although again this capacity is boosted by LABC (Fig 4b). Accuracy in the LABC condition on the most focused (contrasting) test questions is lower than in the Experiment 1 ($\sim 80\%$, depending on the held-out domain), but well above the model trained with random answer candidates ($\sim 60\%$). Interestingly, a model trained on semantically-plausible sets of candidate answers (LABC) performs somewhat worse on test questions whose answers are merely perceptually-plausible than a model trained in that (normal) regime. This deficit can be largely recovered by interleaving random-answer and contrasting candidates during training.

### 3.3.1 EXPERIMENT 3: NOVEL DOMAIN VALUES

Another way in which a domain can be unfamiliar to a network is if it involves attributes whose values have not been observed during training. Since each of the seven (source and target) domains our analogy problems permits 10 values, we can measure *interpolation* by withholding values 1, 3, 5, 7 and 9 and measure *extrapolation* by withholding values 6, 7, 8, 9 and 10. To extrapolate effectively, a model must therefore be able to resolve questions at test time involving lines or shapes that are darker, larger, more-sided or simply more numerous than those in its training questions.

In the case of interpolation, we found that a model trained with random candidates performs very poorly on the more challenging contrasting test questions (Fig 4c 45% vs 93% for LABC), which suggests that models trained in the normal regime overfit to a strategy that bears no resemblance to human-like analogical reasoning. We verified this hypothesis by running an analysis where we presented only the target domain sequence and candidate answers to the model. After a long period of training in the normal regime, this 'source-blind' model achieved 97% accuracy, which confirms that it indeed finds short-cut solutions that do not require analogical mapping. In contrast, the accuracy of the source-blind model in the LBAC condition converged at 32%.

We also found, somewhat surprisingly, that LBAC results in a (modest) improvement in how well models can extrapolate to novel input values (Fig 4c); a model trained on questions with both contrasting and random candidate answers performs significantly better than the normal model on the test questions with contrasting candidate answers (62% vs. 43%), and mantains comparable performance on test questions with random candidate answers (45% vs. 44%).

### 3.3.2 EXPERIMENT 4: MODEL COMPARISON

Finally, we explore the extent to which LABC improves generalistion for different various different network architectures, taking as a guide the models considered in Barrett et al. (2018). We run the **Novel Domain Transfer** experiment with each of these models, trained using both LABC and in the normal training regime. We measure generalisation on a mixed set comprised equally of test questions with semantically-plausible incorrect answer candidates (matching LABC training) and

those with merely perceptually-plausible incorrect answers (matching normal training). All models perform better at novel domain transfer when trained with LABC (Table 1). This confirms that our method does not depend on the use of a specific architecture. Further, the fact that LABC yields better performance than normal training on a balanced, mixed test set shows that it is the most effective way to train models in problem instances where the exact details of test questions may be unknown.

### 3.4 CONCLUSION

These results demonstrate that LABC increases the ability of models to generalize beyond the distribution of their training data. This effect is observed for the prototypical analogical processes involving novel domain mappings and unfamiliar target domains (Experiments 1 & 2). Interestingly, it also results in moderate improvements to how well models extrapolate to perceptual input outside the range of their training experience (Experiment 3).

## 4 EMERGENT RELATIONAL REPRESENTATIONS

To understand the mechanisms that support this generalisation we analysed neural activity in models trained with LABC compared to models that were not. First, we took the RNN hidden state activity just prior to the input of the candidate panel. For a model trained via LABC, we found that these activities clustered according to relation type (e.g. `progression`) more-so than domain (e.g., `shape colour`) (Fig 5a, Table 2). In contrast, for models trained normally the relation-based clusters overlapped to a greater extent (Fig 5b, Table 2). Thus, LABC seems to encourage the model to represent relations more explicitly, which could in turn explain its capacity to generalise by analogy to novel domains.

| Cluster | LABC | Normal Training |
|---|---|---|
| inter-relation dist. | 5.2 ($\pm$ 2.2) | 4.4 ($\pm$ 2.0) |
| inter-domain dist. | 1.8 ($\pm$ 1.8) | 2.0 ($\pm$ 1.8) |

Table 2: **Mean (+SD) distance between clusters of RNN hidden state representations.** LABC seems to promote greater distinctions between different abstract relations in the representation space of models. Since the maximum euclidean distance between two points in a 64 dimensional unit cube is $\sqrt{\sum_i^{64} 1^2} = 8$, the distances that we observe between relational representations are close to the maximum possible.

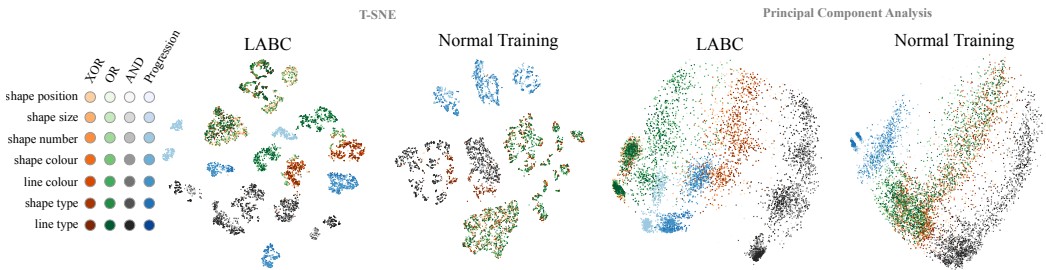

Figure 5: **LABC supports emergent relational representations.** Principle component analysis (PCA) (right) and t-SNE analysis (left) of RNN hidden states. Each dot represents a (64-dimensional) state coloured according to the relation type and domain of the corresponding question.

## 5 SYMBOLIC ANALOGY PROBLEMS

Many of the most important studies of analogy in AI involve symbolic or numerical patterns (Hofstadter, 1996), and various neural models of semantic cognition more generally also operate on discrete, feature-based representations of input concepts (Rogers & McClelland, 2004; Devereux et al., 2018). To verify our findings in this setting, we implemented a symbolic analogy task based on feature-based stimuli. This more controlled domain allows to show that the construction of appropriate incorrect answer candidates can be learned in a proposal model that is trained jointly with

a model that learns to contrast the candidates, widening the potential applications of LABC to task settings where we lack a clear understanding of the underlying abstract relational structure (Sec 5.3).

In our task, inputs are $D$-dimensional vectors $v$ of discrete, integer-valued features (analogous to semantic properties such as *furry* or *carnivorous* in previous work). Across any set $V : v \in V$ of stimuli, each feature dimension then corresponds to a domain (the domains of skin-type or dietary habits in the present example). To simulate a space of abstract relational structures on these domains, we simply take a set $F$ whose elements $f$ are common mathematical functions operating on sets: `MIN`, `MAX`, `ARGMIN`, `ARGMAX`, `SUM` and `RANGE`. Abstract relations in this context can be stated, for instance, as `SUM(1, 2, 3; 6)`. Given a such a function $f$, a set $V$ of stimulus vectors, and a random choice of domain $d$, we can compute a ($D$-dimensional) answer vector $a_{[V,d,f]}$ as the result of applying $f$ to $d$ on $V$ (i.e. executing $f$ on the $d$-th dimension of each $v \in V$). It is now simple to randomly construct an analogy question in this setting. We select a function $f$, source $d_s$ and target $d_t$ domains at random, randomly generate source $V_s$ and target $V_t$ stimuli. We then apply $f$ to $V_s$ on $d_s$ and to $V_t$ on $d_t$ to generate the source and target domain solutions, $a_{[V_s,d_s,f]}$ and $a_{[V_t,d_t,f]}$, respectively. The inputs $V_s$, $d_s$, $a_{[V_s,d_s,f]}$, $V_t$ and $v_t$ are passed to the model, together with a $d \times k$ matrix of alternative answer choices that includes $a_{[V_t,d_t,f]}$ ($k$ is a fixed number of choices, four in the visual analogies presented previously). We then require the model to select which of these alternatives is the true completion of the analogy $a_{[V_t,d_t,f]}$. As in the visual analogy tasks, to resolve such a question the model must detect an abstract relationship in the input domain $d_s$, that explains the connection between the source stimuli $V_s$ and the answer vector $a_{[V_s,d_s,f]}$. Once this achieved, the model must evaluate the function $f$ that describes this relationship on the source domain $d_t$ with sufficient accuracy that it can identify the result of that evaluation $a_{[V_t,d_t,f]}$ in the context of $k$ distracting alternative (incorrect) answers.

We study generalization in this task by restricting the particular ordered domain mappings $d_s \to d_t$ that are observed in the training and test set; for example, aligning structure from domain 3 to domain 1 may be required in the test set, but may have never been seen before in the training set. While we withhold particular alignments (e.g., $3 \to 1$), we ensure that all dimensions are aligned 'out-of' ($3 \to$) and 'into' ($\to 1$) at least once in the training set. Note that this setup is directly analogous to the 'Novel Domain Transfer' experiment in the visual analogy problems (Sec. 3.2).

## 5.1 METHODS

The high level model structure was similar to that of the previous experiments: candidates and their context were processed independently to produce scores, which we put through a softmax and trained with a cross-entropy loss. See appendix 7.2 for further details. Producing appropriate candidate answers $C$ to train by contrasting abstract relational structures (LABC) is straightforward. For a problem involving function $f$, we sample functions $\hat{f} \sim F \setminus \{f\}$ at random and populate $C$ with the $c_{\hat{f}}$, where $c_{\hat{f}} = \hat{f}(V_t)$. In other words, $c_{\hat{f}}$ adheres to *some* relational structure, but just not the structure apparent in the source set. Thus, to determine which candidate is correct, the model necessarily has to first infer the particular relational structure of the source set. Because the implementation of this training regime requires knowledge of the underlying structure of the data (i.e. the space of all functions), we refer to this condition as **LABC-explicit SMT**.

## 5.2 EXPERIMENT 1: NOVEL DOMAIN TRANSFER

We replicated the main findings of the visual analogy task (Experiment 1 S3.2): models trained without LABC performed at just above chance, whereas models trained with LABC achieved accuracies of just under 90%. Note that we tested models with candidates generated using the functions in $F$ not equal to the function used to define the relation in the source set. If instead we were to generate random vectors as candidates, the models could simply learn and embed all possible $f \in F$ into their weights, and choose the correct answer by process of elimination; any candidate that does not satisfy $f(V_t)$ for any $f$ is necessarily an incorrect candidate. This method does not require any analogical reasoning, since the model can outright ignore the relation instantiated in the source set, which is precisely a type of back-door solution characteristic of neural network approaches. These results are intuitive at first glance – a model that cannot use back-door solutions, and instead is required to be more discerning at training time will perform better at test time. However, this intuition is less obvious when testing demands *novel domain transfer*. In other words, it is not obvious that a model

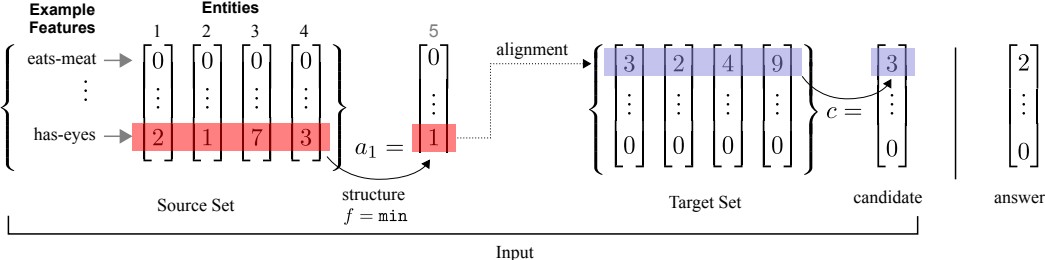

Figure 6: **Structure alignment and mapping on symbolic, numeric data**. In this task a particular structure is implemented as a set-function, which in the depicted example is $f = \min$. For example, the "answer" $a_1$ could denote the minimum size from the sizes in the source symbolic vector set. The model must then evaluate the minimum for the "aligned" domain, which in the depicted case is *intensity*. In this depicted example the candidate does not adhere to this structure, so it would be an incorrect candidate. The correct candidate would look like the answer vector to the right of the image.

that has learned to discern the various functions in $F$ would necessarily be able to flexibly apply the functions in ways never-before-seen, as is demanded in the test set.

## 5.3 EXPERIMENT 2: NOVEL DOMAIN TRANSFER WITH AUTOMATIC LABC METHODS

We explored ways to replicate the results of Experiment 1 without hand-crafting the candidate answers. For our first method, (**LABC-topk**) we uniformly sampled integer values for the $c \in C$ from within some range $(0, 64)$ rather than computing them as $\hat{f}(V_t)$ for some $\hat{f} \in F$. As mentioned previously, such a method should encourage back-door memorization based solutions, since for most candidates, $c \neq \hat{f}(V_t)$ for any $\hat{f} \in F$. To counter this, we randomly generated a set of candidates, performed a forward pass with each, selected the top-$k$ scalar scores produced by the model, and backpropagated gradients *only* through these $k$ candidates. Intuitively, this forces the model to train on only those candidates that are maximally confusing. Thus, we rely on random generation to chose our contrasting candidates, and rely on the model itself to sub-select them from within the randomly generated pool. This method improves performance from chance (25%) to approximately 77%.

It is possible that this top-$k$ method simply exploited random sampling to stumble on the candidates that would have otherwise been hand-crafted. Indeed, for more difficult problems (involving real images, for example) a random generator may not produce anything resembling data from the underlying distribution, making this method less suitable. We thus replicated the top-$k$ experiment, but actively excluded randomly generated candidates that satisfied $c = \hat{f}(V_t)$ for some $\hat{f} \in F$. Performance fell to 43%, confirming this intuition, but interestingly, still greatling improving baseline performance.

| Training method | Test accuracy |
|---|---|
| LABC; explicit SMT | 0.89 |
| LABC; top-k | 0.77 |
| LABC; adversarial | 0.62 |
| Random candidate answers | 0.25 |

Table 3: Test performance on our symbolic analogy in the four different training regimes.

Finally, we considered a method for generating candidates that did not depend on random generation (**LABC-adversarial**), but instead exploited a *generator* model. The generator was identical to the model used to solve the task except its input consisted only of the target set $V_t$, its output was a proposed candidate vector $c$. This candidate vector was then passed to the original model, which solved the analogy problem as before. The generator model was trained to *maximize* the score given by the analogy model; i.e. it was trained to produce maximally confusing candidates. The overall objective therefore resembled a two-player minimax game (Goodfellow et al., 2014):

$$\min_\theta \max_\phi \mathcal{L}(f_\theta(S_1, a_1, S_2, a_2, g_\phi(S_2)))$$

where $f_\theta$ is the analogy model and $g_\phi$ is the candidate proposal model. Using this method to propose candidates improved the model's test performance from chance (25%) to approximately 62%.

These latter experiments show interesting links between LABC, GANs (Goodfellow et al., 2014), and possibly self-play (Silver et al., 2017). Indeed, the latter two approaches may be seen as automated methods that can approximate LABC. For example, in self-play, agents continuously challenge their opponents by proposing maximally challenging data. In the case of analogy, maximally challenging candidates may be those that are semantically-plausible rather than simply perceptually plausible. To our knowledge no prior work has demonstrated the effects of adversarial training regimes on out-of-distribution generalization in a controlled setting like the present context.

## 6 DISCUSSION

Our experiments show that simple neural networks can learn to make analogies with visual and symbolic inputs, but this is critically contingent on the way in which they are trained; during training, the correct answers should be contrasted with alternative incorrect answers that are plausible at the level of relations rather than simple perceptual attributes. This is consistent with the SMT of human analogy-making, which highlights the importance of inter-domain comparison at the level of abstract relational structures. At the same time, in the visual analogy domain, our model reflects the idea of analogy as closely intertwined with perception itself. We find that models that are trained by LABC to reason better by analogy are, perhaps surprisingly, also better able to extrapolate to a wider range of input values. Thus, making better analogies seems connected to the ability of models to perceive and represent their raw experience.

Recent literature has questioned whether neural networks can generalise in systematic ways to data drawn from outside the training distribution (Lake & Baroni, 2018). Our results show that neural networks are not fundamentally limited in this respect. Rather, the capacity needs to be coaxed out through careful learning. The data with which these networks learn, and the manner in which they learn it, are of paramount importance. Such a lesson is not new; indeed, the task of one-shot learning was thought to be difficult, if not impossible to perform using neural networks, but was nonetheless "solved" using appropriate training objectives, models, and optimization innovations (e.g., (Santoro et al., 2016; Finn et al., 2017)). The insights presented here may guide promising, general purpose approaches to obtain similar successes in flexible, generalisable abstract reasoning.

Earlier work on analogical reasoning in AI and cognitive science employed constructed symbolic stimuli or pre-processed perceptual input (Carbonell (1981); Hummel & Holyoak (1997); Hofstadter (1996); Larkey & Love (2003) inter alia; see Gentner & Forbus (2011) for a full review). More recently, Reed et al. (2015) learn an analogy model on top of pre-trained visual embeddings of geometric figures and rendered graphics, while Mikolov et al. (2013) show how analogies can be made via non-parametric operations on vector-spaces of text-based word representations. While the input to our visual analogy model is less naturalistic than these latter cases, this permits clear control over the semantics of training or test data when designing and evaluating hypotheses. Our study is nonetheless the only that we are aware to demonstrates such flexible, generalisable analogy making in neural networks learning end-to-end from raw perception. It is therefore a proof of principle that even very basic neural networks have the potential for strong analogical reasoning and generalization.

As discussed in Sec. 5.3, in many machine-learning contexts it may not be possible to know exactly what a 'good quality' negative example looks like. The experiments there show that, in such cases, we might still achieve notable improvements in generalization via methods that learn to play the role of teacher by presenting alternatives to the main (student) model, as per Shafto et al. (2014). This underlines the fact that, for established learning algorithms involving negative examples such as (noise) contrastive estimation (Smith & Eisner, 2005; Gutmann & Hyvärinen, 2010) or negative sampling (Mikolov et al., 2013), the way in which negative examples are selected can be critical[3]. It may also help to explain the power of methods like self-play (Silver et al., 2016), in which a model is encouraged to continually challenge itself by posing increasingly difficult learning challenges.

**Analogies as the functions of the mind**    To check whether a plate is *on a table* we can look at the space above the table, but to find out whether a picture is on a wall or a person is on a train, the equivalent check would fail. A single *on* function operating in the same way on all input domains could not explain these entirely divergent outcomes of function evaluation. On the other hand, it seems implausible that our cognitive system encodes the knowledge underpinning these apparently

---

[3]See Lazaridou et al. (2015) for a further interesting example of this

distinct applications of the *on* relation in entirely independent representations. The findings of this work argue instead for a different perspective; that a single concept of *on* is indeed exploited in each of the three cases, but that its meaning and representation is sufficiently abstract to permit flexible interaction with, and context-dependent adaptation to, each particular domain of application. If we equate this process with analogy-making, then analogies are something like the functions of the mind. We believe that greater focus on analogy may be critical for replicating human-like cognitive processes, and ultimately human-like intelligent behaviour, in machines. It may now be time to revisit the insights from past waves of AI research on analogy, while bringing to bear the tools, perspectives and computing power of the present day.

## ACKNOWLEDGMENTS

Thanks to Greg Wayne and Jay McClelland for very helpful comments, and to Emilia Santoro, Adam's most important publication to date.

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

## 7 MODEL DETAILS

### 7.1 VISUAL ANALOGY PROBLEMS

The CNN was 4-layers deep, with 32 kernels per layer, each of size $3 \times 3$ with a stride of 2. Thus, each layer downsampled the image by half. Each panel in a question was $80 \times 80$ pixels, and greyscale. The panels were presented one at a time to the CNN to produce 9 total embeddings (3 for the source sequence, 2 for the target sequence, and 4 for each candidate). We then used these embeddings to compile 4 distinct inputs for the RNN. Each input was comprised of the source sequence embeddings, the target sequence embeddings, and a *single* candidate embedding, for a total of 6 embeddings per RNN-input sequence. We passed these independently to the RNN (with 64 hidden units), whose final output was then passed through a linear layer to produce a single scalar. 4 such passes (one for each source-target-candidate sequence) produced 4 scalar scores, denoting the model's evaluation of the suitability of the particular candidate for the analogy problem. Finally, a softmax was computed across the scores to select the model's "answer". We used a cross entropy loss function and the Adam optimizer with a learning rate of $1e^{-4}$.

### 7.2 SYMBOLIC ANALOGY PROBLEMS

A given input consisted of a set of vectors 16-dimensional vectors. This set included 8 vectors comprising $S_1$, one vector $d_1$, 8 vectors comprising $S_2$, and 8 vectors comprising the set of candidate vectors $C$. Vectors were given a single digit binary variable tag to denote whether they were members of the source or target set (augmenting their size to 17-dimensions).

We note that the entity vectors have 0's in their unused dimensions. While this may make the problem easier, this experiment was designed to explicitly test *domain-transfer generalization*, moreso than an ability to discern the domains that need to be considered, by stripping away any difficulties in perception (i.e., in identifying the relevant domains), and seeing if the effect of LABC persists. Thus, at test time the model should have an easy time identifying the relevant dimensions, but it will never have seen the particular transfer from, say, dimension $i$ to dimension $j$. So, even though it may have an easy time identifying and processing each dimension $i$ and $j$, it may be incapable (without LABC) of integrating the information processed from each of these dimensions.

We employed a parallel processing architecture, similar to the visual analogy experiments, with a Relation Network (128 unit, 3 layer MLP with ReLU non-linearities for the $g_\theta$ function and a similar 2-layer MLP for the $f_\phi$ function) replacing the RNN core. Thus, a single model processed $(S_1, d_1, S_2, c_n)$, with $c_n$ being a different candidate vector from $C$ for each parallel pass. The model's output was a single scalar denoting the score assigned to the particular candidate $c_n$ – these scores were then passed through a softmax, and training proceeded using a cross entropy loss function. We used batch sizes of 32 and the Adam optimizer with a learning rate of $3e^{-4}$.

## 8 SUPPLEMENTARY RESULTS

| | | LABC (Train) | Normal (Train) | Mix (Train) | LABC (Contrasting) | Normal (Contrasting) | Mix (Contrasting) | LABC (Normal) | Normal (Normal) | Mix (Normal) |
|---|---|---|---|---|---|---|---|---|---|---|
| Extrapolation | Mean | 0.94 | 0.95 | 0.94 | 0.62 | 0.43 | 0.56 | 0.45 | 0.44 | 0.39 |
| | Std | 0.005 | 0.007 | 0.005 | 0.02 | 0.009 | 0.012 | 0.01 | 0.01 | 0.04 |
| Interpolation | Mean | 0.94 | 0.97 | 0.94 | 0.93 | 0.45 | 0.89 | 0.65 | 0.89 | 0.87 |
| | Std | 0.003 | 0.003 | 0.003 | 0.004 | 0.004 | 0.008 | 0.01 | 0.006 | 0.01 |
| Novel Domain Transfer | Mean | 0.88 | 0.83 | 0.85 | 0.87 | 0.48 | 0.88 | 0.7 | 0.82 | 0.79 |
| | Std | 0.015 | 0.01 | 0.015 | 0.005 | 0.02 | 0.009 | 0.01 | 0.01 | 0.02 |
| Novel Domain (shape colour) | Mean | 0.87 | 0.84 | 0.85 | 0.78 | 0.50 | 0.80 | 0.51 | 0.61 | 0.58 |
| | Std | 0.007 | 0.008 | 0.007 | 0.004 | 0.02 | 0.006 | 0.02 | 0.01 | 0.02 |
| Novel Domain (line type) | Mean | 0.87 | 0.85 | 0.86 | 0.76 | 0.45 | 0.75 | 0.5 | 0.57 | 0.54 |
| | Std | 0.006 | 0.004 | 0.006 | 0.02 | 0.01 | 0.02 | 0.02 | 0.02 | 0.01 |

Table 4: Results for the visual analogy task (RNN Model).

## 9   MODEL COMPARISON DETAILS

Our application of a ResNet-50 processes all nine panels simultaneously (five analogy question panels along with the four multiple choice candidates) as a set of input channels. The Parallel ResNet-50 processes six panels simultaneously as input channels (five analogy question panels along with one multiple choice candidate) to produce a score. Then, similar to the RNN model described above, the candidate with the highest score is chosen by the model. The parallel relation network model also processes six panels simultaneously, using a convnet to obtain panel embeddings and using a relation network (Santoro et al., 2017) for computing a score. For full model architecture details, see the appendix.

Interestingly, the model with strongest generalisation is the parallel relation network, with a particularly high accuracy of $95\%$ on the held out domain-transfer test set. This model was tested on a mixture of multiple choice candidates (that included semantically plausible and perceptually plausible candidates), indicating that models trained with LABC do not over-specialize to problem settings where only semantically plausible candidates are available. We also observe that during normal training, test set performance can oscillate between good solutions and poor solutions, indicated by the high standard deviation in the test set accuracy. These results imply that there are multiple model configurations that have good performance on the training set, but that only some of these configurations have the desired generalisation behaviour on the test set. LABC encourages a model to learn the configurations that generalise at the most abstract semantically-meaningful level, as desired.

We also note that the fact that a model trained in the normal regime performs marinally better than one trained using (normal+)LABC data on test questions involving perceptually-plausible candidates. We believe this may be understood as a symptom of the strong ability of deep learning models to specialize to the exact nature of the problems on which they are trained. The model comparison experiments demonstrate that this negligible but undesirable specialization effect is outweighed by the greater benefits of training with LABC on test questions with semantically-plausible candidates (i.e. those that *require* a higher-level semantic interpretation of the problem). Training with LABC will therefore yield a much higher expected performance, for instance, in cases where the exact details of the test questions is not known.

## 10   FURTHER DISCUSSION AND RELATED WORK

It is interesting to consider to what extent the effects reported in this work can transfer to a wider class of learning and reasoning problems beyond classical analogies. The importance of teaching concepts (to humans or models) by contrasting with negative examples is relatively established in both cognitive science (Shafto et al., 2014; Smith & Gentner, 2014) and educational research (Silver, 2010; Ali, 1981). Our results underline the importance of this principle when training modern neural networks to replicate human-like cognitive processes and reasoning from raw perceptual input. In cases where expert understanding of potential data exists, for instance in the case of active learning with human interaction, it provides a recipe for achieving more robust representations leading to far greater powers of generalization. We should aspire to select as negative examples those examples that are plausible considering the most abstract principles that describe the data.

A further notable property of our trained networks is the fact they can resolve analogies (even those involving with unfamiliar input domains) in a single rollout (forward pass) of a recurrent network. This propensity for fast reasoning has an interesting parallel with the fast and instinctive way in which humans can execute visual analogical reasoning (Morrison et al., 2001; Qiu et al., 2008).

### 10.1   DISTANCE METRIC APPROACHES

LBC shares similarities with distance metric approaches such as the large-margin nearest neighbor classifier (LMNN) (Weinberger & Saul, 2009), the triplet loss (Schroff et al., 2015), and others. In these approaches the goal is to transform inputs such that the distance between input embeddings from the same class is small, while the distance between input embeddings from different classes is large. Given these improved embeddings, classification can proceed using off-the-shelf classification algorithms, such as k-nearest neighbors. We note that these approaches emphasize the form of the loss function and the quality of the resultant input embeddings on subsequent classification. However,

the goal of LBC is not to induce better classification *per se*, as it is in these methods. Instead, the goal is to induce out-of-distribution generalisation by virtue of improved abstract understanding of the underlying problem. It is unclear, for example, whether the embeddings produced by LMNN or the triplet loss are naturally amenable to this kind of generalisation, and as far as we are aware, it has not beed tested. Nonetheless, LBC places a critical focus on the nature, or quality of the data comprising the incorrect classes, and is agnostic to the exact nature of the loss function. Thus, it is possible to use previous approaches (e.g., LMNN or triplet loss, etc.) in conjunction with LBC, which we do not explore. LBC also shares similarities to recent generative adversarial active learning approaches (Zhu & Bento, 2017). However, these approaches do not explicitly point to the effects of the quality of incorrect samples to out-of-distribution generalisation, nor are we aware of any experiments that test abstract generalisation using networks trained with generative samples.

## 11 VISUAL ANALOGY EXAMPLES

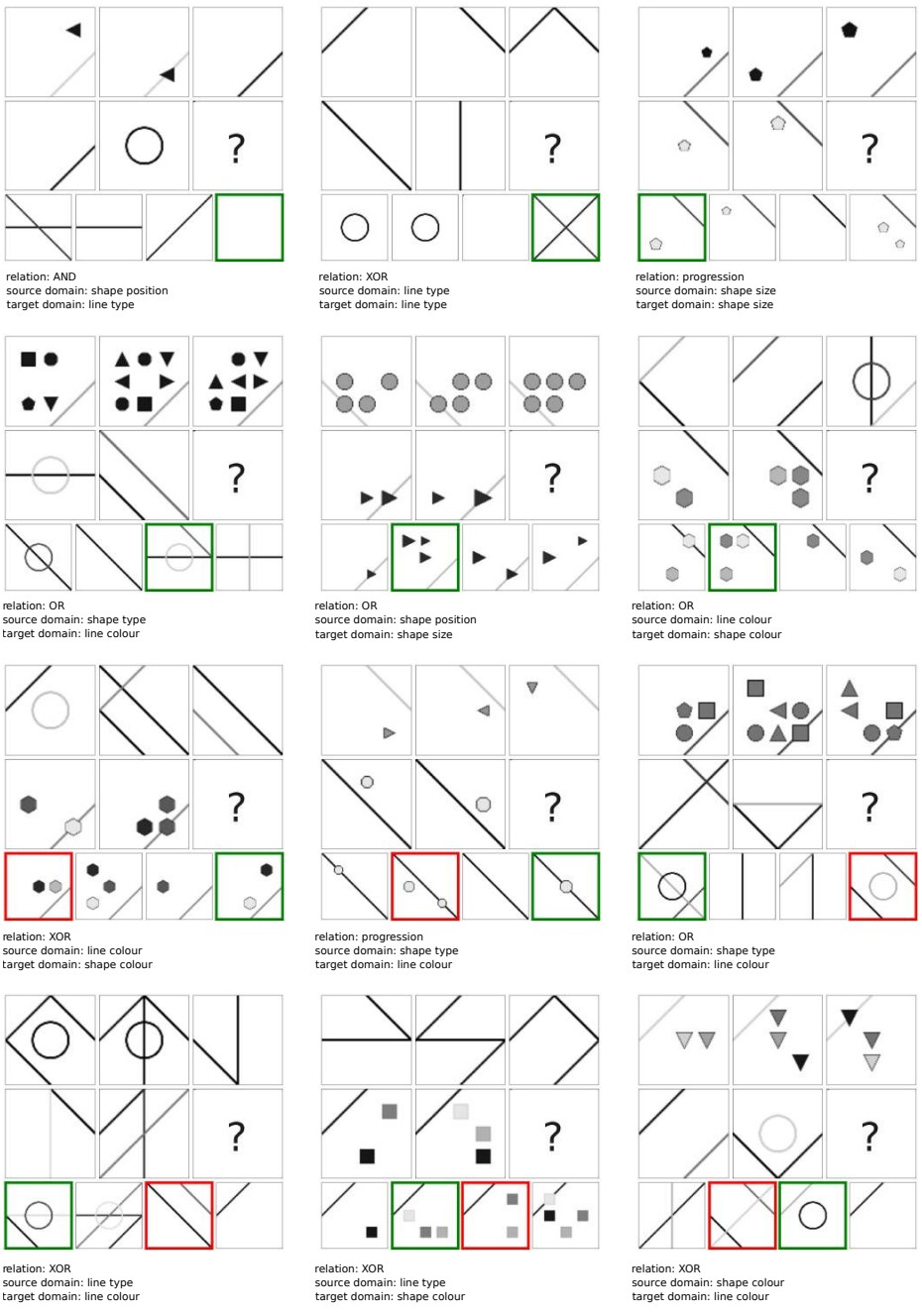

Figure 7: **Examples of visual analogy problems**. These visual analogy examples have been selected from the interpolation test set. The correct multiple choice candidate for each problem is highlighted in green. In the top half, we have randomly chosen examples where our RNN model trained with LABC selects the correct answer. In the bottom half, we have randomly selected examples where our RNN model trained with LABC chooses the incorrect candidates (highlighted in red). The performance of this model on the interpolation test set is 93%.

