# OpenReview forum: "Learning to Make Analogies by Contrasting Abstract Relational Structure"
_ICLR.cc/2019/Conference_

### Official Review · AnonReviewer3 · 2018-11-01
**Intuitive idea for improved training demonstrated powerful in the analogy domain**

**Rating:** 7
**Confidence:** 5

**Review:**

Cons

1.	It’s unclear why LABC produces lower scores than ‘normal’ training on ‘normal’ testing.
2.	The text says nothing I can find to explain why in Fig 5 the ‘entity’ vectors have all 0s except in one dimension, which seems to make the problem considerably easier.
3.	In a sense, there is no cross-domain adaptation required in the symbolic task: min is min, whether it operates on dimension k of the source vectors or dimension j of the target vectors. On the other hand, dimensions are processed independently in the model, as far as I can tell, so there’s no free transfer of learning min on dimension k to knowing min on dimension j. It would be good to comment on this issue.
4.	There seem to be obvious analogies (so to speak) to GANs, and it is very curious that this is not mentioned anywhere that I can see. This is particularly glaring in Sec. 5.3.
5.	The quantitative results are scattered throughout the prose; it would be challenging, but worthwhile, to gather them into an actual table.

Pros

6.	The basic idea (“We should aspire to select as negative examples those examples that are plausible considering the most abstract principles that describe the data”, p. 14) is very intuitive, common-sensical, bordering on obvious. But it is not at all obvious that the idea has as much power as is demonstrated in the experiments. The transfer to novel domain combinations, novel domains, and novel values of dimensions is impressive and surprising.
7.	The result that the proposed training, designed to promote generalization on analogy tasks, also seems to promote improved sensory processing is interesting. Whether it really instantiates the parallel connection argued for by the High-Level Perception view from psychology/philosophy is debatable, but that is itself an interesting connection that the authors should be praised for identifying.
8.	In general, the connection to the cognitive literature is creative and tantalizing and provides good scientific grounding for the work.
9.	The linking to the flexibility of word meanings in the final paragraph pushes the limit of the plausibility of connection to broader cognitive issues, but I’m inclined to indulge the authors for at least bringing up this important and relevant issue.

---

> ### Author Response · Authors · 2018-11-08
> **response to reviewer 3**
>
> Thanks for your review, we're grateful you like the work!
>
> We have given a lot of thought to your point 1. We're running experiments now to mix clever (semantically-plausible) and random (perceptually-plausible) candidates in a more refined way, hoping to train a model that does not degrade at all on test questions involving perceptually-plausible candidates while retaining the strong ability to generalise in the case of semantically-plausible candidates. Would such a result satisfy your reservations here? Having said all this, we don't believe this uncertainty detracts from the fact that a model trained to contrast abstract relational structure generalises more accurately and in a wider-range of out-of-distribution cases than one trained otherwise.
>
> Regarding the entity vectors having all 0’s in the unused dimensions, we agree that this makes the problem easier. This experiment was designed to explicitly test domain-transfer generalization moreso than an ability to discern the domains that need to be considered. The idea was to strip away any difficulties in perception (i.e., in identifying the relevant domains) to see if the effect of LABC persisted. As you note, “dimensions are processed independently in the model...so there’s no free transfer of learning min on dimension k to knowing min on dimension j.” This is indeed the case, and we will clarify this. At test time the model should have an easy time identifying the relevant dimensions, but it will never have seen the particular transfer from dimension i to dimension j. So, even though it may have an easy time identifying and processing each dimension, it may be incapable (without LABC) of integrating the information processed from each of these dimensions, which we demonstrate.
>
> Regarding the link to GANs -- we believe there is a very interesting, potentially deep connection to GAN training, as well as self-play. Please see our reply to R2 for more thoughts here. We think that drawing connections between these methods is a very promising line of future work. We erred on the side of not overclaiming, and not drawing links that we have not rigorously proved, but our minds are definitely oriented in this direction, and we can add some text alluding to these ideas if you believe it necessary.

---

> > ### Comment · AnonReviewer3 · 2018-11-15
> > **reply to response to Rev 3**
> >
> > I agree with all the points in the authors' response to my Review (3).
> > I would indeed be satisfied by "a model that does not degrade at all on test questions involving perceptually-plausible candidates while retaining the strong ability to generalise in the case of semantically-plausible candidates" if you can manage it.
> > About the 0s, the text should state that the vectors have this property, and explain the rationale, i.e., what aspect of the problem the model is addressing.
> > My comment about there being "no free transfer" was actually in reference to the potential criticism that "the cross-domain mapping is trivial, there's nothing to learn there: min maps to min" which as my "no free transfer" comment pointed out, is not actually valid.
> > A mention of GAN is recommended, even as entirely future work, as the omission is GLARING.

---

> ### Author Response · Authors · 2018-11-25
> **Official response to review**
>
> Dear Reviewer Three,
>
> Thank you for your review and very helpful comments.. We note that your principal concerns involved the clarity of exposition of our symbolic experiments and results, our failure to reference GANs and a query (also raised by Reviewer 1) about testing models that were trained using LABC in the 'normal' training condition. To address these concerns, we have:
>
> 1. Added a discussion of the connection between learning by contrasting and GANs in section 5.3
> Clarified the symbolic analogy domain by comprehensively re-writing parts of Section 5 and synthesizing the results in Table 3 (see also our previous response).
>
> 2. Conducted a comparison (see Table 1) that demonstrates clearly the advantage of training with LABC in cases where we do not know in advance whether test questions will involve semantically (and perceptually)-plausible incorrect answers or merely perceptually-plausible answers. See also our response to Reviewer 1 in relation to this.
>
> 3. Added a schematic Figure 1, and re-written Section 3.1 to clarify how analogy problems can be constructed with candidate answers that require the problems to be understood at different degrees of abstract relational structure.
> Added Table 2 (new data) and Table 3 (previously found in main text) to collect quantitative results easier to digest for the reader.
>
> We also draw your attention to the new finding reported in Table 1 that the LABC effect seems to hold for a wide range of well-known visual reasoning neural network architectures, and also the new appendix 7 showing a selection of correct and incorrect test answers given by a model trained with LABC, which gives the reader a better sense of the scope and difficulty of the dataset.
>
> Thanks to your input, we have been able to substantially improve the paper, and hope that you can reevaluate it in light of these extra contributions.

---

> > ### Comment · AnonReviewer3 · 2018-11-27
> > **Strong paper improved**
> >
> > The additional work done in response to reviewers' comments has strengthened the paper significantly. As a cognitive scientist, I place high value on this work. Among the general audience at ICLR, I'm not sure the work will be as appreciated. Thus I'm reluctant to raise the score to an even higher level than the 7 I originally gave. The extra work and clarifications has increased my confidence in that rating though so I'll increase that rating.

---

### Official Review · AnonReviewer2 · 2018-11-02
**The paper describes an approach to train neural networks for analogical reasoning tasks by selecting training instances that force the network to learn relational structure**

**Rating:** 7
**Confidence:** 3

**Review:**

The paper describes an approach to train neural networks for analogical reasoning tasks.

General analogical reasoning is quite a significant milestone in Machine learning. Therefore, the paper tackles an extremely challenging problem.  The paper does a good job of constructing various tasks to show that analogies can be learned in different scenarios which are complex analogy tasks. Specifically, visual analogy and symbolic analogies are considered. The main idea is to choose training examples such that the model is forced to learn the relational structure rather than simply learn superficial features. One weakness is that we need to hand code the training examples to force it to have contrasting relational structure for different tasks. Is this realistic in different problems? That is maybe a limiting factor of this work. An automated method for generating such examples is given, but there is not too much detail on this (5.3). Maybe this needs to be expanded.

Also, is the idea of LABC different from SMT. The novelty may be a bit weak in this aspect. If LABC  can be described in a more general manner, it would help a reader not familiar with the other related work.  Since the baseline comparison is with a very weak method (randomly chosen examples), it is hard to judge the impact of the proposed approach. In summary, I think the paper has nice ideas, particularly, if we can automatically generate examples using LABC. but maybe there is a need to work on better organizing the ideas, more general formulation of LABC and a more convincing experimental evaluation that includes a state-of-the-art method if available

---

> ### Author Response · Authors · 2018-11-08
> **response to reviewer 2**
>
> Thank you for your review!
>
> You mention that one weakness is the need to hand code the training examples such that they have contrasting relational structure. We believe that there may be some deep, important links between LABC, generative adversarial training, and even self-play dynamics. Indeed, the latter two approaches may be seen as automated methods that can approximate LABC. For example, in self-play, agents continuously challenge their opponents by proposing maximally challenging data. In the context of our work, maximally challenging data are analogous to answer choices that are congruent at the level of abstract relational structure, as opposed to simply perceptual structure.
>
> Although we believe there are links to these other methods that may naturally automate candidate generation, no work, to our knowledge, has explored the effects of generative adversarial training or self-play on out-of-distribution generalization, especially in a training setup as we have demonstrated. And so, since this link is as of now just intuitive, we did not want to overclaim in the paper and propose these as natural methods of automating an LABC-like procedure to induce out-of-distribution generalization. We had hoped that our experiment on generative candidate proposals in the symbolic analogy task could point in this direction without overclaiming. Do you think it is appropriate to expand on these points in the paper?
>
> There is a very important point that we’d like to clarify. You note that “[s]ince the baseline comparison is with a very weak method (randomly chosen examples), it is hard to judge the impact of the proposed approach”. We believe this may be a misunderstanding. The baseline comparison is actually quite strong, as we do not choose *truly random* candidates in this condition. Rather, the candidates in this baseline condition are *perceptually plausible* given the visuals of the context panels (i.e. they are always taken from the target domain). For example, if the context panels contain 1 circle, then 2 circles, then a possible answer choice would have 4 circles. A possible answer choice would *not* contain something as truly random as various colored lines; if the training questions had such truly random candidates we think it is obvious that the model would learn very little of interest at all - it would learn the most superficial ability to match the visuals of the question and the answer, and yield no generalisation whatsoever. If you would like, we can run an experiment to demonstrate this, and will update the text to reflect our procedure for generating baseline candidates.
>
> Finally, regarding LABC and SMT, there are indeed some crucial differences. First, SMT is a psychological description of a human phenomena. It describes how humans tend to make analogies by mapping relational structures from one domain to another. LABC, on the other hand, is a *training* method. LABC contends that if a model is coaxed into *learning* to map relational structure, then it will be better at making analogies, as evidenced by out-of-distribution generalization. And indeed, our results show that this may be the case. Thank you for pointing out that these ideas are not clear -- we will try to update the description of LABC to better situate it in the previous literature.

---

> > ### Comment · AnonReviewer2 · 2018-11-20
> > **Reading author response**
> >
> > I am happy with the author response. Thanks for the clarification regarding experiments..I think the experiments section is a little more clear to me in terms of the baselines used. Also I think "applying" the idea of SMT into a concrete learning algorithm seems to be a good contribution.

---

> ### Author Response · Authors · 2018-11-25
> **Official response to review**
>
> Dear Reviewer Two,
>
> Thank you for your review. Your comments have allowed us to make various additional contributions, resulting in substantial improvements to the paper. We note that your principal concern with the work was with the impact and novelty of the proposed approach, and have addressed this concern with the following contributions:
>
> 1. We have added a clear discussion of the connection between learning by contrasting and other methods, and how it might be applicable in different contexts (Section 5, end).
>
> 2. We have added a schematic Figure 2 that highlight the status of the candidate answers in the 'normal' training regime; it should now be clear that there is a hierarchy of possible choices from entirely random to perceptually plausible to semantically plausible. We have also made this much more clear in the text, Section 3.1, paragraph 2. If training with entirely random candidate answers it would be trivial to identify the correct answer as that belonging to the target domain of the question (but the model would not learn much of interest in this case). The comparisons in this paper are instead between a training regime where candidates are randomly chosen from the target domain (perceptually-plausible) and those in the target domain that comply with some other abstract relation (perceptually and semantically-plausible).
>
> 3. We have added an additional experiment (see Table 1), where we show that the advantage of LABC applies to various other well-known visual reasoning models.
>
> 4. We have added an Appendix 7 showing a selection of correct and incorrect test answers given by a model trained with LABC to give the reader a better sense of the diversity of the data and where models fail.
>
> Please also note our previous comment about the connections between SMT (a psychological theory) and LABC (an approach to training neural networks)
>
> We believe that these additional contributions, made following your recommendations, have made the paper considerably stronger.  We hope that you can reconsider your view in light of this improvement.

---

### Official Review · AnonReviewer1 · 2018-11-05
**Interesting direction and nice discussions, but evaluation and analysis are not strong enough**

**Rating:** 6
**Confidence:** 3

**Review:**

This work investigates the ability of a neural network to learn analogy. They showed that a simple neural network is able to solve analogy problems with image or abstract input, given that the training data is selected to contrast abstract relational structures.

The paper is relatively well-written with rich discussions. Some details about the experiments are missing like how many examples are used for training and testing. It is also important to show how much variations are in the dataset, and there should be some external baselines like those proposed in (Barrett et al, 2018).

Although the performance is relatively high, some error analysis will provide more insights into what the neural network is missing and if it makes mistakes similar to human.

Section 4 claims that “For a model trained via LABC, we found that these activities clustered according to relation type (e.g. progression ) more-so than domain”. However, it is unclear whether Figure 4 can support this. Some quantitive measure should help, for example, the average distance within the clusters between clustering based on relation type and domain.

The novelty of the proposed approach is limited. The difference between the proposed method and baseline in performance seems to be a result of whether there is a difference between train and test setting. For example, if trained in “contrasting” will have better test performance on “contrasting” but worse on “normal” and vice versa.

The problem is very interesting and the discussion is extensive. However, the proposed approach isn’t very novel and the evaluation and analysis should be improved to provide a stronger support.

Barrett, David GT, et al. "Measuring abstract reasoning in neural networks." arXiv preprint arXiv:1807.04225 (2018).

------

Score updated after reading authors' response.

---

> ### Author Response · Authors · 2018-11-08
> **response to reviewer 1**
>
> Thanks for the review -- we’re glad you find the problem interesting!
>
> You are right that we could have included more details of the visual analogy dataset to give a sense of its variety and scope. The training set contained 600,000 samples, 10,000 for validation, and 100,000 for testing. Regarding the variation in the dataset: are you referring to the number of possible questions? While the number of domains and relations is small, the combinatorics of the puzzles (born out through the choices for source and target domains, the relations, the values of the attributes, etc.) ensures that the dataset is highly variable. For example, a quick back-of-the-envelope calculation shows that there are nearly 8 billion possible analogy questions (source domain + target domain) that the generator could produce. Moreover, variability in the possible incorrect choices further increases the number of possible questions to an astronomical amount. I hope this gives some sense of the scope of the dataset: we will add a table detailing these statistics to the appendix.
>
> We agree that it would be interesting to see how the model fails and whether the ways are similar to the ways in which humans fail. We’re performing this analysis now, and will be able to share specific failure cases with you soon. Additionally, we are testing a wider range of existing models models. We’d like to note, though, that the appropriate baselines are comparisons *within* model-type; that is, using normal or LABC training on the same model. Our choice of model (an RNN -- not even an LSTM!) was deliberately as simple as possible so as to emphasize the effect of the training method; we agree, however, that it would be useful to corroborate the effect we observed using other well-known (and closer to state-of-the-art) architectures.
>
> Regarding quantitative measures for the clustering analysis. We performed this analysis and found that the inter-class cluster means are greater in LABC compared to normal training (5.2 vs. 4.4), confirming the intuition that can be gleaned from the figure. We will soon compile further analyses with results of new experiments into a new comment addressed to all reviewers.
>
> Would you mind clarifying your opinion that the method is not novel? As far as we are aware, there is no prior work that has analyzed the effects of negative examples on out-of-distribution generalization. Indeed, there is very little work at all addressing out-of-distribution generalization in abstract relational spaces. The closest work that we are aware is the triplet loss in computer vision, and some recent work in generative adversarial training. However, this work did not study the effects of the training method on out-of-distribution generalization, as we have. If you can provide specific examples of similar work, we'll try to explain where this study differs.

---

> ### Author Response · Authors · 2018-11-25
> **Official response to review**
>
> Dear Reviewer One,
>
> Thank you for your review. Your comments have allowed us to make substantial improvements to the paper. We note that your principal concern with the work was the strength of the analysis and evaluation, and have addressed this with the following additional contributions:
>
> 1. To make the evaluation stronger, we have added a set of results for a range of known visual reasoning architectures (Table 1), including an implementation of the relation-network model of Barrett et al. The effect of LABC is maintained in each case.
>
> 2. To make the analysis stronger, we have added quantitative analysis of the representations learned by the models in the two conditions (now Table 2) , to complement the qualitative analysis and plots that we produced previously.
>
> 3. To further improve the analysis, we have added and discussed examples of particular questions where the model fails in the two conditions (Appendix, Figure 7).
>
> 4. To make the scope of the task clearer to the reader, we have added statistics of the training and test sets in different conditions (Section 3.1, final paragraph).
>
> As you point out, when testing the model in the baseline condition (perceptually plausible candidates), there is a small drop in performance for a model trained by contrasting (semantically plausible candidates) compared with one trained in the baseline condition. While a complete understanding of (non-convex) optimisation and generalisation in multi-dimensional spaces of data is beyond the scope of this work, this effect can presumably be understood as a symptom of the models' ability to specialise to the questions they are trained on. However, it is not correct to equate this drop with the gains observed when testing on questions with contrasting answers.
>
> 5. To make this final point explicit, we conducted the futher experiments reported in (new) Table 1. These comparisons make it clear that, unless we have specific information that questions involving semantically-plausible candidates will not be found in the training set, all other things being equal better performance can be expected when training the model with semantically-plausible candidates (i.e. LABC).
>
>
> We believe that these additional contributions make the evidence of the generality of the observed effect when learning analogies by contrasting even stronger, and therefore considerable improve the paper. We hope that you can reconsider your view in light of this improvement.

---

> > ### Comment · AnonReviewer1 · 2018-11-27
> > **Updated score based on author response**
> >
> > Thanks for the response. The update has improved the paper significantly. To clarify my concern about novelty, it seems that the change in the accuracy comes from whether the training and testing distribution match or not. If the distribution matches (train on "contrast", test on "contrast", or train on "normal", test on "normal"), you will get better accuracy than when the distribution doesn't match (train on "normal", test on "contrast", or train on "contrast", test on "normal"). Nevertheless, I think the analysis and discussion is interesting, so I will update my rating accordingly.

---

### Author Response · Authors · 2018-10-01
**Error in caption of figure 1**

The caption to Figure 1 refers to an old caption. It should read: In this analogy puzzle, the model must identify a relation
(Progression ) on a particular domain ( shape quantity ) in the source sequence (top), and apply it to a different domain ( line color) in order to find the candidate answer panel that correctly completes target sequence (bottom). Apologies - we will fix up asap.

---

### Public Comment · ~Douglas_Blank1 · 2019-05-08
**Related work**

For a comparison of making visual and structural analogies in an early neural network, please see:

Blank, D. 1997. Learning to See Analogies: A Connectionist Exploration. PhD. thesis. Indiana University. https://repository.brynmawr.edu/cgi/viewcontent.cgi?article=1077&context=compsci_pubs

Although this work is very different in many aspects, there are some similarities and surely lies in related work.

---

### Meta-Review · Area_Chair1 · 2018-11-05
**Interesting and relevant work, improved in revision**

**Confidence:** 3
**Recommendation:** Accept (Poster)

**Metareview:**


pros:
- The paper is well-written and includes a lot of interesting connections to cog sci (though see specific clarity concerns)
- The tasks considered (visual and symbolic) provide a nice opportunity to study analogy making in different settings.

cons:
- There was some concerns about baselines and novelty that I think the authors have largely addressed in revision

This is an intriguing paper and an exciting direction and I think it merits acceptance.